# Light-Driven Depolymerization of Cellulosic Biomass into Hydrocarbons

**DOI:** 10.3390/polym15183671

**Published:** 2023-09-06

**Authors:** Arvind Negi, Kavindra Kumar Kesari

**Affiliations:** 1Department of Bioproduct and Biosystems, Aalto University, 02150 Espoo, Finland; 2Department of Applied Physics, School of Science, Aalto University, 02150 Espoo, Finland; 3Research and Development Cell, Lovely Professional University, Phagwara 144411, Punjab, India

**Keywords:** biomass, lignocellulose, photothermal oxidation, cellulose, hemicellulose, C5-hydrocarbons, C6-hydrocarbons, sustainable chemistry

## Abstract

Cellulose and hemicellulose are the main constituents of lignocellulosic biomass. Chemical derivatization of lignocellulosic biomass leads to a range of C5 and C6 organic compounds. These C5 and C6 compounds are valuable precursors (or fine chemicals) for developing sustainable chemical processes. Therefore, depolymerization of cellulose and hemicellulose is essential, leading to the development of various materials that have applications in biomaterial industries. However, most depolymerized processes for cellulose have limited success because of its structural quality: crystallinity, high hydrogen-bond networking, and mild solubility in organic and water. As a result, various chemical treatments, acidic (mineral or solid acids) and photocatalysis, have developed. One of the significant shortcomings of acidic treatment is that the requirement for high temperatures increases the commercial end cost (energy) and hampers product selectivity. For example, a catalyst with prolonged exposure to high temperatures damages the catalyst surface over time; therefore, it cannot be used for iterative cycles. Photocatalysts provide ample application to overcome such flaws as they do not require high temperatures to perform efficient catalysis. Various photocatalysts have shown efficient cellulosic biomass conversion into its C6 and C5 hydrocarbons and the production of hydrogen (as a green energy component). For example, TiO_2_-based photocatalysts are the most studied for biomass valorization. Herein, we discussed the feasibility of a photocatalyst with application to cellulosic biomass hydrolysis.

## 1. Introduction

Biomass conversion has become an attractive field for producing fine chemicals and fuels. Biomass contains lignin as an aromatic organic component, while cellulose and hemicellulose are sugar polysaccharide components, as shown in Figure 1 and Table 1. These components constitute ~60 wt% of lignocellulosic biomass, representing a source for producing renewable hydrocarbons, as shown in Figure 1. Hydrolysis of cellulosic biomass leads to various types of fine chemicals (C5 and C6-hydrocarbons), which have high significance for pharmaceutical and biopolymer industries. Improvements in processing methods, especially those involving physiochemical parameters, have demonstrated product selectivity. Therefore, such methods in recent years enjoyed their adoption at various pilot and upscale processes for large-scale commercializing of these chemicals. 

## 2. Challenges in Cellulose Hydrolysis: Influence of Structure Function

### Structure of Cellulose

As the most abundant biopolymer and with wide application in various fields, cellulose processing is considered challenging. The shortcoming of cellulose chemical processing lies within its structure. Structurally, it is a linear, crystalline homopolymer of anhydroglucose units with a high degree of polymerization (ranging from 300 to 10,000) [8]. These units, linked together by *β*-1,4-glycosidic bonds, have a chair conformation, as shown in Figure 2. The -OH groups in these units are in equatorial positions, displaying a high network of intermolecular hydrogen bonds (involving adjacent anhydroglucose units); therefore, a strong intermolecular hydrogen bonding (H-bond) is one of the key characteristics, packing the cellulose chains in such a way to form a crystal structure that poses a challenge to mild chemical acid penetration (or treatments). This additional H-bonding produces an ambiguous solubility profile of cellulose with common organic solvents and lesser miscibility with water. Therefore, cellulose reactions are usually heterogeneous, and derived products dissolve in the reaction media, exhibiting insolubility and requiring additional processing to achieve higher hydrolysis of cellulose. 

## 3. Methods of Cellulosic Biomass Conversion

Notably, the global energy demand increase in recent years requires alternative fuel materials, which has led to heavy investment in production and newer processing routes from various multinational companies working in energy fields. Cellulosic-based biomass processing fuels are considered carbon-neutral feedstock, where carbon dioxide (CO_2_) is produced during its transformation and can be recaptured (CO_2_ fixation) into lignocellulose biomass. Therefore, cellulose could serve as an alternative feedstock that can decrease our dependence on fossil fuels and help mitigate CO_2_-driven climate change. For example, ionic-liquid-based treatments for bioethanol using ultrasonication with ionic liquid ([TEA][HSO_4_]) from wheat straw [7].

Therefore, there are various ways to describe the conversion of cellulosic biomass, but broadly, there are three categories: biocatalysts (the use of enzymes; cellulase is the most used enzyme), hydrothermal processes (which mainly include acid treatments), and photocatalysis (less in use, but recent years have shown tremendous growth in these methods); a generalized representation is provided in Figure 3. 

### 3.1. Enzymatic Processes for Converting Cellulosic Biomass

Various biocatalysts and their recombinant variants were developed in this direction to achieve a higher selectivity in product formation, as shown in Table 2. Because of enzyme stereospecificity and the biocatalytic nature, they specifically hydrolyze the *β*-1,4-glycosidic bonds and require lesser assistance from external factors (such as high temperatures and pressure), therefore decreasing the chances of degradation of fine chemicals or end-products (from the cellulosic biomass). In this direction, most of the work has been performed on the cellulase enzyme, reasonably because of its natural availability and cellulose biomass selectivity (as a substrate).

The enzymatic transformation (or microbial fermentation) of lignocellulosic biomass has been widely used to produce biofuels and fine chemicals [21,22,23]. In an industry process, before implicating the enzymatic hydrolysis on the lignocellulosic biomass, pretreatment is a prerequisite to enhance the efficiency of the enzymatic process [24,25]. As cellulosic biomass (for example, lignocellulose) has intrinsic resistance against hydrolysis, pretreatment (s) of biomass is commonly practiced and gains a reasonable yield from enzymatic hydrolysis. Such pretreatments not only improve the yields of final products but also assist in reducing enzyme loading (quantitatively). Different classes of enzymes (and their presences) are involved in the cellulase system, targeting specific structures (or metabolites) leading to complex biosynthetic pathways of different ratios of biomass components. Therefore, choosing biomass (cellulosic) with appropriate compositions of components for a specific enzyme is one of the common strategies employed in pilot-to-medium-level industries. Reports highlighting maximum sugar content (percentage) on pre-treated biomass require high cellulase loading, making it a high-cost operative component in the whole valorization process [24,26,27]. To address such issues, one could adjust the enzyme formulation to improve the efficiency of enzymatic hydrolysis. For example, adding an additive enzyme (*β*-glucosidase) to mitigate the negative feedback inhibition from its own produced final products or intermediates is one of the interesting approaches (for example, cellobiose, an intermediate that strongly inhibits the activity of cellulase) [28,29]. Therefore, strategies involving synergistic roles of enzymes, enzyme loading, and supplementation of *β*-glucosidase are continuously accessed for futuristic advancement in the enzymatic hydrolysis of cellulosic biomass. Such limitations can be mitigated by introducing a suitable pretreatment method before the enzyme hydrolysis step. In an interesting study, pretreatment methods were employed on Dacotah switchgrass to evaluate dependent enzymatic hydrolysis on these pretreatments, as shown in Table 3. This study measured the xylose and glucose yields at both stages (pretreatment and enzyme hydrolysis). The pretreatment methods include dilute sulfuric acid, sulfur dioxide-impregnated steam explosion, liquid hot water, ammonia fiber expansion, soaking in aqueous ammonia, and lime treatment. It has been observed that lower pH pretreatments (such as diluted sulfuric acid, sulfur dioxide steam explosion, and liquid hot water) solubilized higher amounts of glucose than higher pH pretreatment methods (ammonia fiber expansion, soaking in aqueous ammonia and lime). Furthermore, pretreatments demonstrated total glucose yields comparatively higher than untreated cellulosic biomass (even at lower cellulase loadings), proving the significance of pretreatment before inducing the enzymatic hydrolysis. In conclusion, lime and sulfur dioxide steam-explosion-based pretreatments were found to be closer to the maximum yield possible of glucose (60.6%), while the pretreatment method (soaking in aqueous ammonia) had a lower yield of 39.9%.

Various commercial activities are undergoing in the direction of cellulosic ethanol development [22,30,31,32]. However, these industries are encountering challenges in upscaling the pretreatment method. At the same time, enzyme loading still costs too much, reflecting a high increase in the final product cost, ultimately hampering the broad adoptability of such technologies in the competitive fuel market [22,30]. Such limitations not only narrow the application of lignocellulose biomass valorization in biofuel production but clearly show a need for more technological advancement. One strategy that can improve the process is to use concentrated biomass (also “high-solids”) for the enzymatic treatment stage, also called “high-solids enzymatic hydrolysis” [33,34,35]. The term “high solids” indicates the involvement of insoluble solid content in a slurry form with low to no free water [36,37], resulting in a more efficient upscaling of final products and reducing the operation cost (energy consumption) [38,39]. Additionally, it reduces the overall operation volumes, thereby decreasing the wastewater and subsequent distilling (in the case of cellulosic ethanol production) [40,41]. One of the downsides of high-solids enzymatic hydrolysis is that it prompts technical difficulties in continuous operations, as loss of free water from biomass substrate significantly enhances the viscosity, influencing the uniformity mixing (unequal mass and heat transfer in the process) and thereby reducing the enzyme efficiency in the early stage of hydrolysis (liquefaction stage) [42]. Therefore, attaining a continuous flow of operation or reproducibility of yield is a bit subjective, known as the “high-solids effect” [43]. As the biomass was composed of various components, measuring the relative influence of those on the resultant process and final yields could be accessed accurately and relatively limited. 

### 3.2. Acidic Treatments for Converting Cellulosic Biomass

Cellulose chemical composition of anhydroglucose units (linked to *β*-1,4 glycosidic bonds) and H-bonding (intra- and intermolecular H-bonding) leads to its limited solubility in water and is not ideal for initiating any processing (or its derivatization), therefore requiring additional auxiliaries. In the above section, we discussed the enzymatic methods (cellulase and its recombinants); such methods are selective but are not cost-effective on a large scale. Therefore, the adoption of enzymatic methods on an industrial scale is limited and requires chemical treatments, which are cheaper. One example is acid catalysis, another promising strategy for hydrolyzing cellulose into fine chemicals. Mineral acids mediated up to 80% hydrolysis but with the application of high temperatures; therefore, degraded products (moderate percentages) can be seen in such processes. However, these methods produce many acidic components (or side products) and facilitate at high temperatures, posing a significant aquatic ecotoxicity and hazard to in-line facility instrumentation. Compared to acidic hydrolysis of cellulose, enzymatic hydrolysis undoubtedly has environmental benefits but has over 10% higher cost in implementation. Therefore, significant effort has been put forward to develop solid acid catalysts that illustrate high affinity towards cellulose, with increasing interest in addressing these issues. This includes metal oxides, metal phosphates, polymer acid resins, sulfonated carbonaceous acids, hetero-poly acids, and H-form zeolites. The mechanism of solid catalysts for cellulose conversion into fine chemicals can differ based on the surface topology of the employed catalyst (overall surface and active site), solvent, and pretreatments for substrate activation. However, the limiting factor for the success of solid acid catalyst is directly dependent on its mass diffusion with crystalline cellulose, which can be enhanced by adopted treatment technology (microwave digestion and other mechanochemistry methods such as ball milling) and a sustainable increase in physical parameters such as high temperature and pressure. However, on a large scale, these parameters are unsuitable for catalyst stability and suffer from low product selectivity and unreasonably high energy consumption, making the process costly. These issues have encouraged researchers to develop effective chemical auxiliaries and catalysts.

However, cellulose crystallinity and intensive H-bonding networking impart resistance to its hydrolysis, which can therefore be achieved with a harsh acidic medium at high temperature but is flawed by commonly producing end products such as hydromethylfurfual (HMF), levunillinic acid (LeA), and humins (as shown in Figure 4). 

### 3.3. Photocatalytic Conversion of Cellulosic Biomass

The catalyst provides an interface to the reactants or intermediates (formed during the reaction), either through its surface (adsorption or weaker electrostatic interactions) or intermolecular bonding, resulting in a decrease in activation energy (E_a1_ in an uncatalyzed reaction mode) to a catalyzed reaction mode (activation energies: E_a2_ and E_a3_), as shown in Figure 5a. Such a reduction in activation energies (uncatalyzed to catalyzed reaction modes) demonstrates the successful implementation of these thermocatalysis approaches. Such changes in activation energies lead to the control of reaction kinetics (length of the reaction to completion) and product selectivity. In various instances, these also led to regioselective products (with the synthesis of a stereoisomer). However, these approaches are highly dependent on high temperatures to achieve a reaction’s activation barrier rapidly but often result in degrading the organic components at such temperatures. High temperature also leads to heating the overall system, and sensitive organic components (carrying multiple functional groups) are degraded to multiple side-products. Therefore, implementing thermocatalysis can be cost-effective but results in other side-stream organic components (C5-hydrocarbons: humins) and decreases end-product purity (or target products transformed from cellulosic biomass valorization). Such pyrolysis reactions often require a temperature above 300 °C [44], while hydrogenation requires 150 °C or above [45]. Therefore, additional parameters were evaluated to address the issues arising from high-temperature-dependent biomass conversion reactions (such as product selectivity and the degradation of organic materials). One of the promising catalysis strategies that have witnessed advancement in recent years is photocatalysis, which uses light energy, thereby decreasing the dependence on high temperature or pressure (in some cases) [46,47]. The preliminary step for photocatalysis requires a light source from which the catalyst in the reaction obtains energy (photocatalyst). Light adsorption with an appropriate wavelength by the catalyst transformed it in its active form (excited photocatalyst state) to interact with the reactant (or intermediates), thereby decreasing the activation energy of the uncatalyzed reaction (as shown in Figure 5b). Such photocatalysis can be activated under mild physical factors (temperature and pressure). Furthermore, a photocatalyst or light-activating catalyst is much more suited for biomass valorization as various organic components (multifunctional) are present, therefore providing more product selectivity and lowering the degradation of products into unnecessary components (for example, humins production at high temperatures from lignocellulose valorization). Interestingly, reported the biomass valorization has undergone reduction–oxidation (redox) reactions, requiring a stoichiometric amount of oxidant (photogenerated reactive species, for example, H_2_O_2_) and reductant (photo-excited charge carriers—for example, H_2_).

Reports indicate the role of localized surface plasmon resonance (LSPR) in applying photocatalysis for a non-redox reaction. An LSPR-induced temperature increment produces a rapid, selective nanoscale effect around the surface (of a molecule or catalyst); therefore, its scope can be enlarged by altering the intensity and wavelength of the light exposed.

Ke et al. from the School of Chemistry at the Queensland University of Technology, Brisbane, Australia, reported the first example of photocatalytic hydrolysis of cellulose to glucose using a plasmonic nanostructure (composed of TiO_2_ nanofibre-supported H-form Y-zeolites decorates with Au-NPs and Au-HYT) under visible-light irradiation [49]. They utilized new nanocatalysts that might be doped with plasmon metal—for example, Au-NPs (which could act as “antennas” to absorb light of particular wavelengths or UV-Vis radiations). The localized surface plasmon resonance (LSPR) effect is a physical absorption of light, which produces a boost in the electromagnetic field of a localized surface up to 1.0 × 10^2^–1.0 × 10^5^ times. In recent years, there has been an advancement in using plasmonic metals for photochemical catalysis. Ke et al. used Au-NPs to substantially improve zeolite photocatalysis with higher selectivity (product). Furthermore, other plasmonic metal catalysts were also evaluated: Au-NP-decorated zeolite Y on TiO_2_ NFs (Au-YT), TiO_2_-supported Y zeolite (YT), TiO_2_-supported HY zeolite (HYT), acid-treatedTiO_2_ NFs (HT), and HY zeolites, as shown in Table 4 and Figure 6.

Initially, the authors dissolved the cellulose with 1-ethyl-3-methylidiazolium chloride solution, later followed by the addition of a photocatalyst and water under light irradiation. No products were observed with 100% water, but a relative mixture of water: ionic liquid (1:9) was found to be optimum, indicating the role of water in cellulose solubility and swelling. Plasmonic metal nanoparticles were loaded onto the solid catalyst to absorb visible light and improve catalyst activity and product selectivity. This is feasible because light irradiation can intensify zeolite extra-framework ions’ strongly polarized electrostatic fields [50,51]. Cellulose conversion proceeded effectively on these catalysts, even at mild reactive conditions, as shown in Table 4. The comparative study of loaded versus unloaded catalysts in dark and light conditions and a significant conversion rate of glucose and HMF indicated that Au-HYT has strong photocatalytic activity, as shown in Figure 6. Heating at 140 °C for 16 h under visible light, a conversion yield of 58.7% (glucose, 48%; HMF, 10.6%) was attained. Mechanistically, it is hypothesized that these photocatalytic differences in conversion yields are mainly due to the LSPR effect of Au-NPs, and to some extent, the acid strength supported the polarized electric field of zeolites owing to the LSPR effect of plasmonic Au NPs, as shown in Figure 7.

Li and co-workers developed a plasmonic metal catalyst (Ir/HY catalyst) for efficient hydrolysis of a *β*-1,4-glycosidic bond of cellobiose with 99% conversion using continuous illumination of visible light from a 300 W xenon lamp (equipped with 420 nm cut-off filter) within a temperature not exceeding 100 °C [52]. Similar to the previous studies by Ke and co-workers, they also found no photoactivity for HY zeolite. However, with Ir-nanoparticles, the photocatalytic yield was 2.8-folds higher under visible light irradiation than under dark conditions. In these experiments, the exclusive formation of glucose and 5-HMF as products indicated that the Ir-photocatalysis is highly selective towards the *β*-1,4-glycosidic bond cleavage rather than typical C-C bond cleavage. Furthermore, to access the other possible Ir-catalyst support, SiO_2_ and P25 were evaluated, which were found to be inactive (<1%) in the cellobiose hydrolysis under visible light irradiation.

To expand the scope of support, increasing Si/Al ratios of HY zeolites led to improved activity in the dark. Additionally, the Ir/HY3 catalyst (with Si/Al = 11) led to higher hydrolysis at 90 °C, and this catalyst behavior is also comparable with the thermal reaction. Amberlyst-15 resin support showed the highest activity (at temperature = 70 °C). The photocatalytic trends were observed: Ir/Amberlyst-15 > Ir/HY3 (Si/Al = 11) > Ir/HY-450 (Si/Al = 7) > Ir/HY2 (Si/Al = 5.2) > Ir/SiO_2_. Furthermore, these experiments also indicated that the acid density of supports is directly proportional to the catalytic activity and plays a vital role in the hydrolysis of cellobiose. The metals (nanoscale) transform the light energy into heat energy (an example of the LSPR effect), increasing the temperature localized to the surface when compared to the conventional thermocatalysis method, therefore producing a rapid increase in the temperature of that localized surface area (of the catalyst or targeted substrate), facilitating the catalytic efficiency (photocatalysis). Although the postulated mechanisms of photocatalysis by these metal nanoparticles, as suggested, seem different, as shown in Figure 7B,C, similar photoexcitation phenomenons were expected to occur in these metal–zeolite structures to accelerate the catalytic transformation of cellulose into C6-hydrocarbon (glucose). 

To understand the mechanism, further studies were carried out to reveal the photocatalytic degradation of cellulosic biomass by photogenerated holes or other reactive species. In the photo-reforming process, the photocatalyst redox activity reduces H^+^ to H_2_ and subsequently oxidizes the organic compound. Therefore, in the photo-reforming of cellulosic biomass, when performed for H_2_ production (using semiconductors), simultaneously the photoexcited electrons reduce water into H_2_ and holes (or system-generated reactive species like radicals OH˙) expected to cellulosic biomass oxidization [53]. Carbon dioxide is one of the primary C1 sideline products released during the cellulosic biomass valorization, along with intermediatory C6 hydrocarbons (as glucose) and C1 hydrocarbon (formic acid) using platinum absorbed on titanium oxide support (as a photocatalyst under ultraviolet irradiation) [54]. Interestingly, similar photocatalysis (platinum absorbed on titanium oxide support) in sulphuric acid (0.6 M) under light irradiation leads to cellulose and the formation of hydroxymethylfurfural (12.8% as the primary product) [55], while without light irradiation, glucose was found to be a major product. Such observations indicate the role of the photogenerated holes or other system-generated radical species or reactive species in the oxidative hydrolysis of intermediatory products (especially glucose units) to other lower carbon-containing organic compounds (humins) or CO_2_.

It would be more informative if future experiments studied HMF photocatalysis. Importantly, preliminary studies showed the synthesis of hydroxymethylfurfural (as the primary product) during the valorization of cellulosic biomass in a highly concentrated zinc chloride solution (66%) using titian oxide under UV irradiation [56]. Unlike acid-mediated oxidative hydrolysis of the *β*-1,4-glycosidic bond, the radical or reactive species generated in the system (during photocatalysis) facilitate the cleavage of the *β*-1,4-glycosidic bond using a radical mechanism with the involvement of photogenerated holes on titanium oxide. Additionally, zinc chloride (as a Lewis acid) seems to be isomerizing the C6 hydrocarbons (glucose → fructose), thereby producing the hydroxymethyl furfural via dehydration.

A higher ratio of oxygen: carbon in C_5_–C_6_ hydrocarbons derived from cellulosic biomass is significantly important (as organic precursors) for large-scale platforms [57]. However, we already discussed various examples of how difficult it is to gain high product selectivity when performing the valorization of cellulosic biomass. As discussed earlier, the multiple functional abundances in the cellulosic biomass are one of the key areas; therefore, more sophisticated catalysis is required, which will lead to the development of photocatalytic processes which focus mainly on three types of reaction category: (a) oxidation hydrolysis of -OH and -CHO groups of the cellulosic biomass; (b) carbon–carbon cleavage; and (c) isomerization of C6-hydrocarbon (glucose into fructose), followed by its dehydration into hydromethylfurfural or the mixture of any of two or all three reactions, as shown in Figure 8.

The application of semiconductor catalysts as photocatalysts was a successful strategy when performing cellulosic-biomass valorization [58]. Various species (reactive in nature) can be generated when a semiconductor catalyst is irradiated with an appropriate wavelength of light in the presence of oxygen or water. Hydroxyl radicals can be produced, either H_2_O (oxidation) by photogenerated holes or oxygen reduction (after multiple reactions by photogenerated electrons). Other reactive oxygen species, such as hydrogen peroxide and superoxides radicals (O_2_˙^−^), could be produced with higher amounts of free radical propagation. The reactions usually occur in situ to produce these reactive oxygen species, as shown in Figure 9. During direct degradation with reactive oxygen species, oxidative degradation may occur directly in the photogenerated holes, indicating that the valence band potential must be comparatively more positive than that of the oxidation reaction, as shown in Figure 9.

Titanium oxide displayed efficient photocatalytic conversion rates for gluconic acid and glucaric acid from glucose [59,60,61,62] due to its high oxidative potential (3.2 eV on photogenerated holes of titanium oxide versus standard hydrogen electrode, while that of the formed hydroxyl radicals has 2.8 eV versus standard hydrogen electrode) [63], making them strong oxidants. These acids (gluconic acid and glucaric acid) resulting from C6-hydrocarbons photocatalysis are quite commonly known for their application in cosmetic and biopharmaceutical industries. Being strong oxidants, their oxidization can lead to the valorization of intermediate C5 and C6 hydrocarbons further into other C1 hydrocarbons (usually CO_2_), thereby having lower selectivity towards products (or organic acids). Using the sonication-induced sol-gel method, nanostructured titan oxide was prepared and demonstrated substantially selective conversion (50%) of glucose (11%) with 50% selectivity for glucaric and gluconic acids compared to conventional titanium oxide (P25) [59]. Such observation might be expected from the fact the nanostructure titanium oxide has a lower affinity for these organic acids, therefore facilitating the desorption of both acids from the titanium oxide (nanostructured), abrogating any further oxidation [59]. Interestingly, zeolite Y-supported titanium oxide exhibited preferential (~70%) conversion of glucose (15%) into organic acids under UV-Vis light compared to titanium oxide (P-25), indicating the desorption of gluconic acid and glucaric acid (organic acids) facilitated by the negatively charged zeolite framework [60]. Ultrasonic-mediated impregnation of titanium oxide/silicon oxide or titanium oxide/zeolite-Y with transition metal cations (M^+^) might improve the photocatalytic conversion of C6/C5-hydrocarbons (primarily glucose) into organic acids [61]; chromium-doped titanium oxide/zeolite-Y demonstrated photocatalytic conversion (7%) into organic acids (with 87% efficiency). Importantly, iron-doped titanium oxide/zeolite-Y revealed photocatalytic preferential efficiency of 94% towards organic acids with a conversion rate of 7% [62], indicating that the doping of the M+ cations of chromium and iron into titanium oxide lattice reduced the optical bandgap, reciprocating a decrease in the oxidation potential of titanium oxide and thereby preventing further oxidative reactions of organic acids. 

Photocatalysts based on Metallothioporphyrzines (MPz) also showed valorization of C6 hydrocarbons (derived from cellulosic biomass) to organic acids [64,65,66]. Metallothioporphyrzines have a macrocyclic core bearing sulfur atoms in the macrocyclic periphery, known to absorb visible light because of long conjugated systems, and therefore tend to produce different coloring compounds (organometallic compounds) depending on the type of embedded metalcore. Upon irradiation (wavelength in the visible range), metallothioporphyrzines perform oxidative reactions with peroxides or oxygen. For example, H-ZSM-5-supported FePz [66], ZnO-supported CoPz [64], and Tin(IV) oxide-supported FePz [65] with peroxide under visible light irradiation could oxidize glucose, while Tin(IV) oxide-supported FePz showed the highest photocatalytic activity with preferential selectivity for organic acids (52.2%) with conversion of glucose (34.2%) [65].

Zhou et al. from the Beijing National Laboratory for Molecular Sciences, the Chinese Academy of Sciences (Beijing, China), exploited the photocatalysis of titanium oxide-supported gold nanoparticles, where various cellulosic biomass hydrocarbons (including glucose, xylose, ethanol, 2-furaldehyde, furfural alcohol, and HMF) were used [67]. After 96 h of completion under UV/visible irradiation, the yield conversion from glucose and xylose to gluconic acid and xylonic acid reached more than 95%, respectively, indicating the plasmonic gold nanoparticles’ deceiving role in such photocatalysis. Upon vis-light irradiation, the Local Surface Plasma Resonance-induced electrons on the gold nanoparticles were injected into the conduction band of titanium oxide, which was later involved in oxygen activation, thereby producing enough oxygen species reactive to perform selective glucose and xylose oxidation to their respective organic acids, as shown in Figure 10. Additionally, sodium bicarbonate could have prevented the oxidative species (generated under UV-light irradiation) from oxidizing the organic acids [67]. 

Omri et al. investigated three gold-based photocatalysts using sunlight irradiation (AM 1.5G, 100 mW cm^−2^), Au/TiO_2_, Au/CeO_2_, and Au/Al_2_O_3_, for selective oxidation of sugars and other oligosaccharides into corresponding sodium aldonates [68]. They also studied the peroxide used as an oxidant and electron scavenger in the presence of 0.003–0.006 mol % of gold in basic conditions. Further to their investigation, they found the highest photocatalytic efficiency (TOF > 750,000 h^−1^) with maximum yield and purity (>95%). Comparatively, Au/CeO_2_ was found to be more effective with greater LSPR and efficient separation of photogenerated electrons and holes [68]. The selectivity of products could be because of the involvement of aqueous sodium hydroxide that prevented oxidizing species from oxidizing the in situ organic acids [68].

Photocatalysis in anaerobic conditions can lead to chemical degradation (C-C cleavage) of monosaccharides into much simpler forms. For example, Chong et al. from the Dalian Institute of Chemical Physics (the Chinese Academy of Sciences, Dalian, China) reported the acid–base free photocatalytic conversion of glucose to value-added chemicals (arabinose and erythrose) and H_2_ under mild aqueous conditions. Interestingly, the authors found the highest selective conversion for erythrose (65%) and arabinose (91%) using rutile TiO_2_, providing the first evidence that *α*-scission (C_1_–C_2_ cleavage) of glucose was facilitated by photocatalysis to yield arabinose in an aqueous medium. Electron paramagnetic resonance (EPR) spectroscopy and monitoring of glucose reactions were used to understand the mechanism of photocatalytic conversion. In catalytic conversions, selectivities are reported against the irradiation interval of time, where prolonged irradiation increases the glucose conversion rate (demonstrating the photocatalyst’s key role in these conversions) while product selectivity shifts from arabinose to erythrose. The titanium oxide (TiO_2_) samples were prepared using the modified hydrothermal method [69], while co-catalysts were loaded (using the in situ photo-deposition method) in various concentrations (0.2 wt% Rh (or Pt, Pd) or 1.0 wt% Cu, Ni) with the respective forms (RhCl_3_ (or H_2_PtCl_6_·6H_2_O, PdCl_2_, CuCl_2_, or Ni(NO_3_)_2_) before the reaction proceeded (glucose conversion reaction). 

### 3.4. Cellulosic-Biomass-Derived Hydrogen Production

According to the 2016 report, global fossil-based fuel consumption reached a staggering 1100 barrels (nearly 10^5^ L per second) [53,70], which cannot be managed with low-cost fuel for an extended period of decades [71] (or with clean energy) and, therefore, could irreversibly damage the environment (ecosystem) [72]. Hydrogen-based fuel has shown to be a promising alternative due to the advantages of no carbon footprint, which means one gallon of hydrogen fuel emits zero carbon dioxide when combusted [73,74,75]. The design and implementation of hydrogen fuel-based engines are gaining significant interest worldwide in various areas (for example, automobile and aviation industries [76,77,78]). 

However, technologies based on the traditional light-driven catalytic production of hydrogen are plagued with low quantum yields of water splitting, primarily due to the thermodynamic barrier of the oxygen evolution reaction (OER, ΔE⁰ = −1.23 V), the presence of reverse reactions, and rapid recombination of photogenerated charge carriers [79,80]. Additionally, the lack of efficient storage technology, production lines, and low-cost sources is still a significant challenge to its commercial success [81,82,83]. The most common way to produce hydrogen is by reforming/gasifying fossil-derived coal or oil [84]. While processing via biomass conversion seems the most economical route to produce sustainable hydrogen, as it is commonly employed, the cellulosic chemical transformation itself has complications using a high temperature of ≥750 °C, leading to organic components of cellulosic biomass decomposing into hydrogen along with other non-relevant gases, such as carbon monoxide, carbon dioxide, and methane [85,86]. However, seeing the disadvantages of thermal-based chemical processing, much work has been completed to improve the selectivity and efficiency of such biomass conversion, leading to the adoption of photothermal applications. Such incorporation lowers the energy cost as solar energy represents an inexpensive energy source and, therefore, is feasible for commercialization. Such photo reforming of biomass-derived chemicals (biomass photo-reforming) uses light-driven water splitting for hydrogen production. It has shown promising results, as reported in previous literature [87,88,89,90], where it essentially requires a photocatalyst to generate holes to oxidize the cellulosic biomass and exploit the resultant electrons to reduce the aqueous protons (H^+^) to hydrogen [91]. Implementation of such photocatalytic biomass reforming requires an efficient technology that exploits biomass (lignocellulosic biomass or other cellulosic biomass) as a reductive substrate to substitute OER for hole (h^+^) consumption; therefore, cellulosic biomass acts as a hole scavenger, providing a continuous supply of electrons for hydrogen production. Importantly, technological advancements have been made in exploiting photocatalysis, which, upon photoexcitation, resulted in producing holes (h^+^), oxidizing biomass, and subsequently reducing the H^+^ (aqueous protons) using electrons (e^−^) to produce hydrogen. Meanwhile, using cellulosic biomass, methanol, glucose, and glycerol are the most valuable substrates for such applications. Contrary to the oxygen-evolution reaction, the resultant energy remains neutral for photocatalytic-driven biomass-reforming reactions, as they consume highly abundant photons and lead to the formation of value-added products [91,92,93,94]. 

Cellulosic biomass refining requires acidic hydrolysis (or enzymatic/pyrolysis), which is expensive and inefficient in selective substrates [91]. To facilitate commercial success, sustainable hydrogen production (or methods) requires direct reforming of cellulosic biomass comparable to the thermochemical processes [95,96], which is challenging to achieve. However, using non-scalable noble-metal catalysts (such as platinum and ruthenium (IV) oxide (RuO_2_)) loaded over a UV-absorbing titanium oxide surface demonstrates direct photo-reforming of cellulosic biomass into hydrogen [54,97,98]. The Erwin Reisner research group from the Department of Chemistry, the University of Cambridge, Cambridge, the United Kingdom (Christian Doppler Laboratory for Sustainable SynGas Chemistry) reported a mild photo-reforming method of cellulosic biomass to produce hydrogen using cadmium sulfide quantum dots (QDs)-based photocatalysis [53]. Cadmium sulfide is a yellow-colored powder (inorganic pigment, CI pigment yellow 37), has two crystalline forms, and is a direct band gap semiconductor (gap 2.42 eV). Additionally, the proximity of its band gap near the visible light wavelengths provides it with color properties. Its conductivity is enhanced when irradiated, while in combination with a p-type semiconductor, it can become a primary core of a photovoltaic cell (for example, cadmium sulfide/copper(I) sulfide solar cell) and, doping with an activator (Cu^1+^) and coactivators (Al^3+^) under electron beam excitation cadmium sulfide luminance (called as “Cathodoluminescence”), can enable its application as “phosphor” [99]. The distinctive properties of cadmium sulfide make it an interesting choice for the authors to choose it for direct photo-reforming of cellulosic biomass [53]. Comparatively, the potential of conduction band cadmium sulfide is −0.5 eV than the normal hydrogen potential, demonstrating enough potential to reduce a proton, as reported in various photocatalytic conversion rates to hydrogen [100]. Notably, the valence band of cadmium sulfide is +1.9 V (when compared to a normal hydrogen electrode), which is enough for cellulosic biomass oxidation, but can initiate the photocorrosion of the catalyst itself (photo-oxidation of its sulfide), limiting the usability of cadmium sulfide [100]. The authors address these shortcomings by implementing highly alkaline conditions that initiate the formation of cadmium hydroxide/cadmium oxide (Cd(OH)_2_/CdO) on the cadmium sulfide surface. The resultant photocatalytic system (CdS/CdOx based quantum dots) efficiently performs the catalysis (by avoiding the photocorrosion of cadmium sulfide), while light-driven hydrogen is produced by oxidizing the unprocessed cellulosic substrates. At a highly alkaline pH, cellulosic biomass tends to solubilize easily, thereby promoting inter cooperativity between the in situ formation of an active form photocatalyst while increasing in the solubilized cellulosic substrate, ultimately enhancing the kinetics rate of hydrogen formation [53]. Various strategies with the optimized structure of photocatalysts or photocatalytic systems were employed to improve the efficiency of the photo-reforming of cellulosic biomass for hydrogen production. Some reviews that provide further information include those on the implication of photocatalysts in the production of hydrogen from H_2_O [101], photo-reforming of organics for oxidation reactions [58,102,103,104], cooperative semiconductor-based photocatalytic coupling of oxidative organics (such as CxHx, ROH, ROR, RNH_2_) to value-added chemicals along with hydrogen production [105], and photocatalysts (optimized structure or system) for cellulosic biomass (lignocellulose) for value-added chemical and hydrogen production [88]. 

## 4. Conclusions

In conclusion, the valorization of cellulosic biomass poses many challenges to the material chemist (or researchers working in allied fields) due to the poor physicochemical properties of components present in such biomass. Biocatalysis methods mainly involve applying cellulase enzymes (or recombinant forms) with different hydrothermal conditions, which has high selectivity but eventually a high cost, limiting its widespread commercial application. The application of catalysts (with sustained high temperatures) provides a choice (alternative) to biocatalysts when compared to their overall cost but also leads to the degradation of products (because of such high temperatures). Eventually, high temperatures (used along with a catalyst) produce multiple activation energies (or intermediates), resulting in overall non-selective products. Therefore, catalyst (mineral or solid acids)-based processes surely increase the conversion rate but have low product selectivity. Another shortcoming of such an approach is that applying high temperatures also affects the catalytic efficiency of the catalyst when used in iterative cycles. Such limitations are thought to be overcome by photocatalysis, as photocatalysis can occur at lower temperatures or milder conditions because it utilizes photon energy rather than thermal energy and, therefore, does not produce degraded end products and can be used for iterative cycles. Furthermore, these mild conditions do not aggravate the kinetic energy of the molecular bonds and, therefore, create more selective cleaving of certain low-energy bonds to attain the desired products (or end chemicals). For example, titanium oxide and cadmium sulfide break the *β*−O−4 and *β*-1,4-glycosidic bonds in cellulosic biomass [106]. Various catalysts, especially TiO_2_-based photocatalysts, have been shown to achieve higher conversion rates of cellulosic biomass in producing C5 or C6 (xylose, glucose) or their organic acids with reasonable photocatalytic efficiency. Other than TiO_2_-based photocatalysts, other metal oxides (as photocatalysts) have been used to perform photocatalysis of biomass. One prominent study showed that gold-based photocatalysts successfully converted the C6 hydrocarbons (or oligosaccharides) into the oxidized form with around 95% purity. Nevertheless, these results have indicated the possibility of achieving different value-added products directly from cellulosic biomass using photocatalysis.

## Figures and Tables

**Figure 1 polymers-15-03671-f001:**
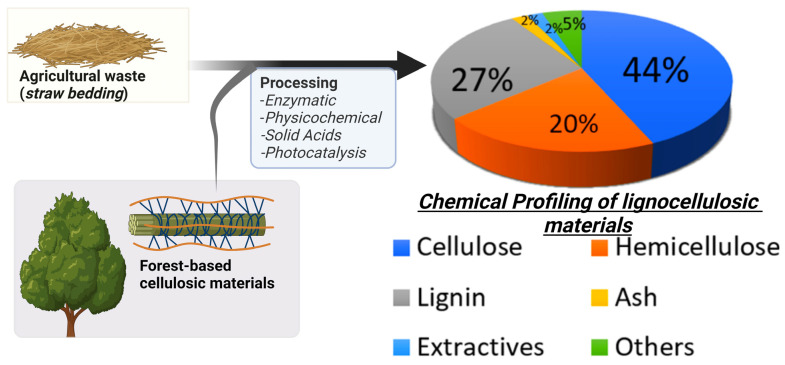
Typical chemical distribution of cellulosic biomass.

**Figure 2 polymers-15-03671-f002:**
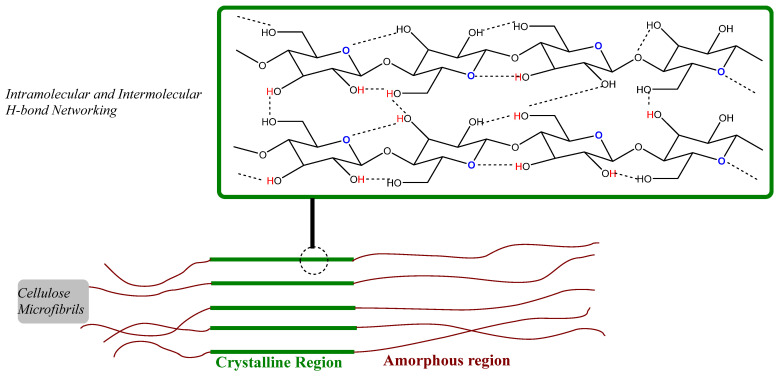
Cellulose structure, defining its crystalline region and amorphous region. Additional H-bonds can be seen as part of intramolecular and intermolecular structures, abrogating the cellulose’s chemical sensitivity.

**Figure 3 polymers-15-03671-f003:**
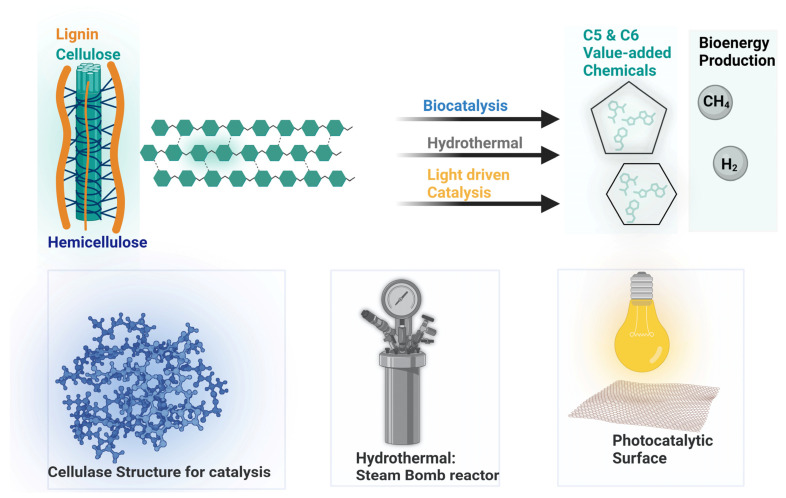
Schematic representation of wood processing into the organic components (C5 and C6 hydrocarbons) using various methods (biocatalysts, hydrothermal catalysis, and photocatalysis).

**Figure 4 polymers-15-03671-f004:**
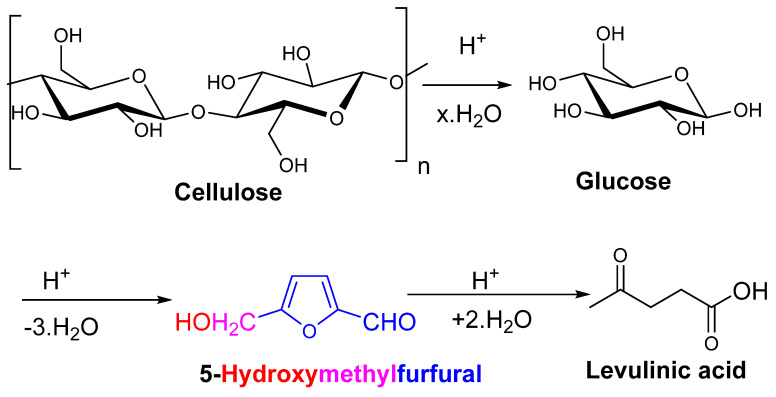
Harsh acidic medium at high temperatures leads to products such as HMF, LeA, and humins.

**Figure 5 polymers-15-03671-f005:**
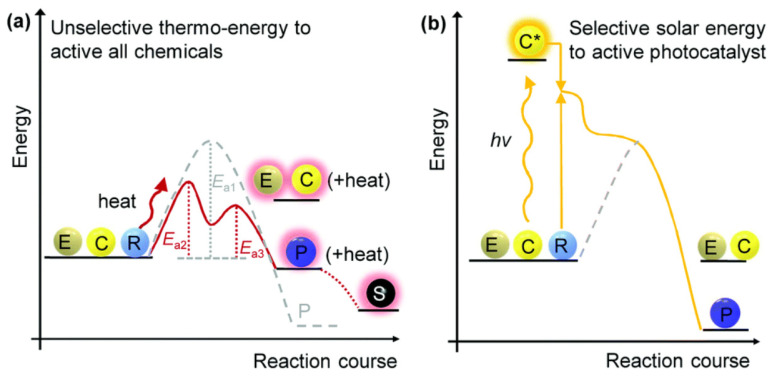
(**a**) Reaction kinetic mechanisms of thermocatalysis show reactants’ step-by-step initiation and activation with the help of catalysts, exhibiting various activation energies (E_a1_, E_a2_, and E_a3_). E_a1_ represents the uncatalyzed reaction mode, while E_a2_ and E_a3_ show catalyzed reaction modes, exhibiting the role of the catalyst in overcoming the activating energy. However, showing more than two activation energies can lead to the formation of more than one intermediate, which could form more than one product, making the reaction non-selective. (**b**) Reaction kinetic mechanisms of photocatalysis show the light energy initially absorbed by the catalyst, making it excited to interact with the reactant under mild conditions, not producing multiple activation energy levels (modes), thereby illustrating the advantage over thermocatalysis in product selectivity. Reproduced with permission from Wu et al. Copyright 2023 RSC [48].

**Figure 6 polymers-15-03671-f006:**
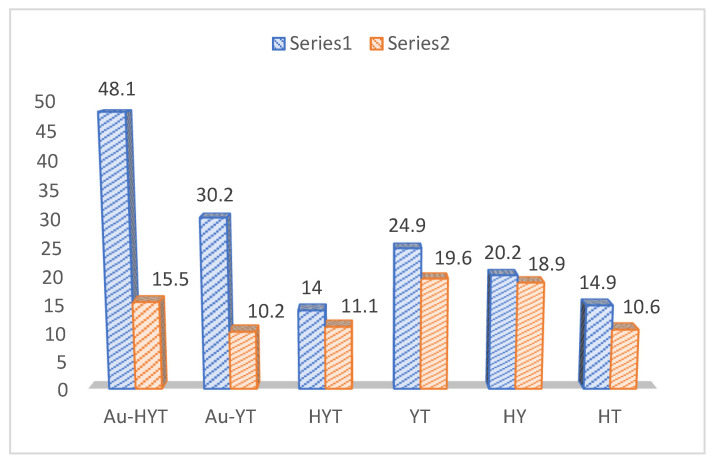
Photocatalytic conversion of cellulosic biomass using photocatalyst (catalysts Au-HYT, Au-YT, HYT, YT, HY, HT), where series 1 represents light ON, while series 2 represents lights OFF.

**Figure 7 polymers-15-03671-f007:**
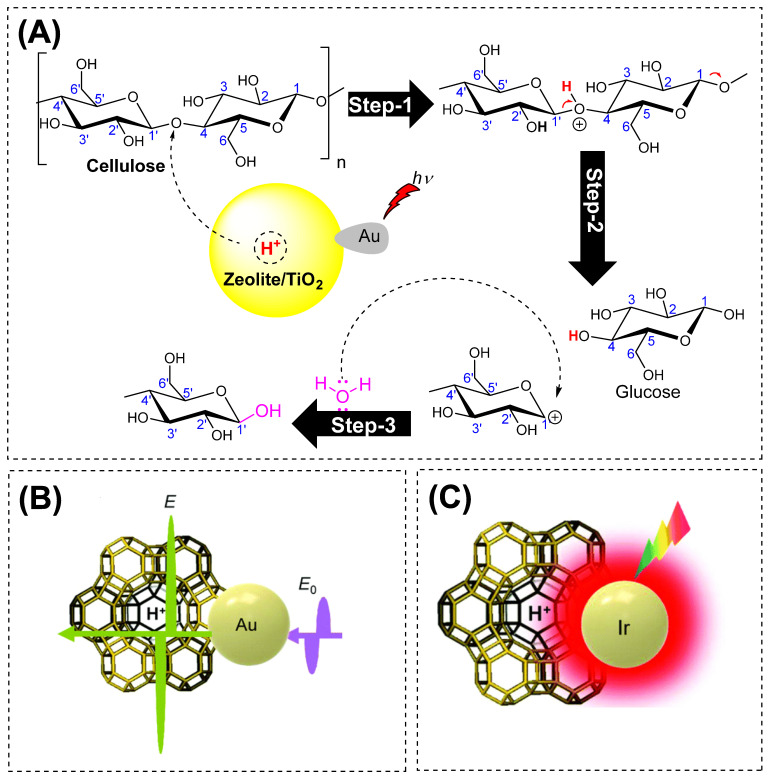
(**A**) The putative photocatalysis mechanism by Au converts the cellulosic biomass into glucose (C6-hydrocarbon) under light irradiation. (**B**) A proposed light-promoted mechanism for Au–zeolite composites. (**C**) A proposed light-promoted mechanism for Ir–zeolite composites [52]. Figure (**B**,**C**) were reproduced with permission from Wu et al. Copyright 2023 RSC [48].

**Figure 8 polymers-15-03671-f008:**
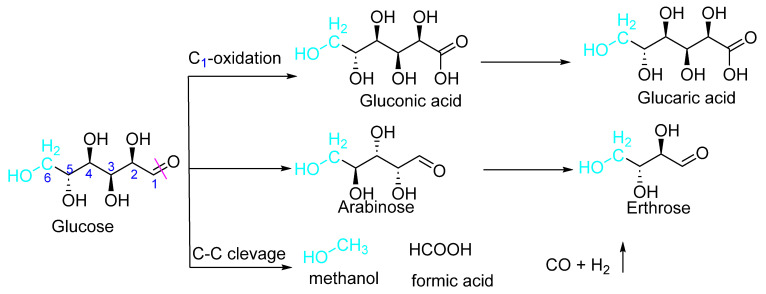
Photocatalysis of C5 and C6-cellulosic biomass-derived hydrocarbons.

**Figure 9 polymers-15-03671-f009:**
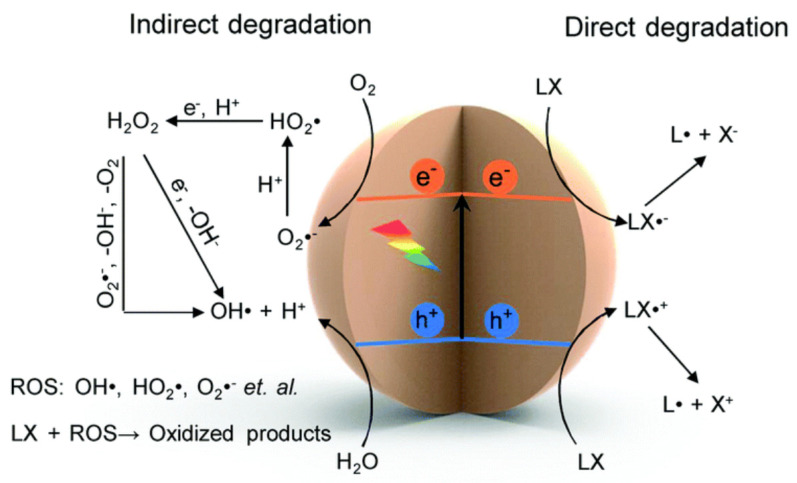
Representation of photocatalysis mechanism of semiconductor catalyst in the presence of oxygen and water. Reproduced with permission from Wu et al. Copyright 2023 RSC [48].

**Figure 10 polymers-15-03671-f010:**
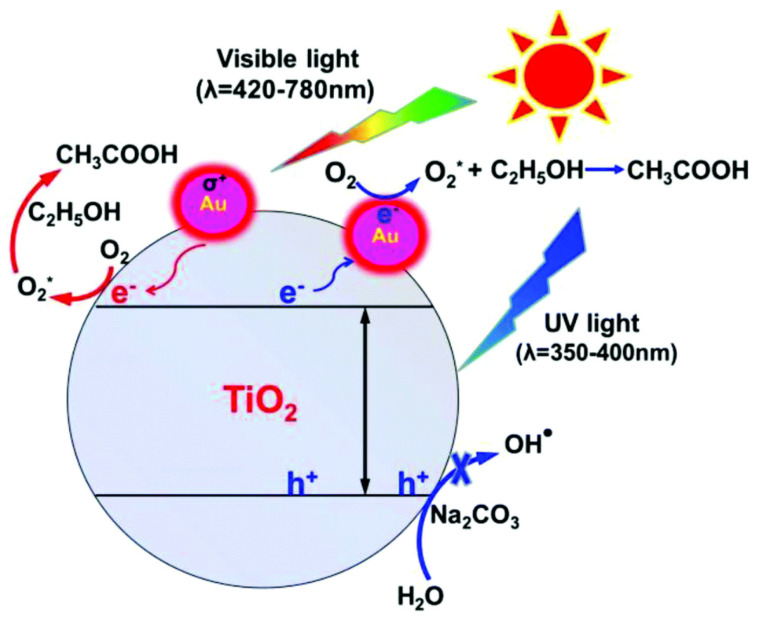
Photocatalytic oxidation of hydrocarbons to organic acids with the application of Au/TiO_2_ in aqueous sodium hydroxide solution using UV-/Vis-light irradiation. Reproduced with permission of Royal Society of Chemistry, Copyright 2017 [67].

**Table 1 polymers-15-03671-t001:** Chemical distribution of cellulosic biomass from various sources (data from Sun et al. [1,2,3,4,5]).

Lignocellulosic Materials	Cellulose (%)	Hemicellulose (%)	Lignin (%)
Hardwoods stem	40–55	24–40	18–25
Softwoods stem	45–50	25–35	25–35
Nutshells	25–30	25–30	30–40
Corn cobs	45	35	15
Grasses	25–40	35–50	10–30
Paper	85–99	0	0–15
Oil palm empty fruit bunches ^a^	NA	24–35	18–35
Wheat straw ^b^	33–38	26–32	17–18
Sorted refuse	60	20	20
Leaves	15–20	80–85	0
Cotton seed fibers	80–95	5–20	0
Newspaper waste	40–55	25–40	18–30
Waste papers from chemical pulps	60–70	10–20	5–10
Primary wastewater solids	8–15	NA	24–29
Swine waste	6.0	28	NA
Solid cattle manure	1.6–4.7	1.4–3.3	2.7–5.7
Coastal Bermuda grass	25	35.7	6.4
Switchgrass	45	31.4	12.0

^a^ [6], ^b^ ash (6–8%) [7].

**Table 2 polymers-15-03671-t002:** Biocatalytic methods for wood and agricultural waste (as cellulosic biomass).

Biomass Source	Hydrothermal (HT) Pretreatments	Biocatalytic Hydrolysis Phase	Outcome	Ref.
*Tamarix ramosissima* bark (Salt cedar)	Hot compressed water (HCW) pretreatment: Deionized H_2_O (10% *w*/*v* of dry solid mass) at 100–200 °C in a batch-type reactor	Cellulase loading 35 FPU/g	Enzymatic saccharification of HCW pre-treated cellulose-rich biomass yield 88% hydrolysis, 3–4 times higher than untreated method	[1,9]
Starch	Hot compressed water (HCW) pretreat-mint in the presence of CO_2_ ^a^ at 180–235 °C for 15 min	Fermentation: *Escherichia coli* HD701 cultured (inoculate 10 μL of sample equiv. to 0.001% inoculum). cell concentration measured by optical density using a conversion factor; OD6001 = 0.482 g dry weight. L^−1^. 0.5–1 mL of cell suspension added to make up the 1 g L^−1^ dry weight concentration	Addition of CO_2_ into HCW method increases sugars yields (~1.5-fold compared to a N_2_ control at 250 °C for 15 min) from cellulose-enriched biomass and decreases yields of organic acids (also act as fermentation inhibitors, such as 5-hydroxymethylfurfural and furfural) in lignocellulose hydrolysates. Activated carbon-treated hydrolysates exhibited higher hydrogen production than untreated, exhibiting the efficiency of activated carbon treatments in removing organic acids (which act as inhibitors) for *Escherichia coli* HD701.	[10]
Sorghum bagasse	Microwave digestion equipment with IR sensor: microwave power (700 W), liquor to solid ratio (6.7 g g^−1^) at 160–210 °C (3–30 min)	Enzymes derived from wild-type strain F3 of *Fusarium oxysporum* and *Neurospora crassa* DSM 1129	Cellulases from *Fusarium oxysporum* demonstrated an extended depolymerization of cellulose (Yield of glucose ~60%) higher than cellulases from *Neurospora crassa* (could be due to relatively higher cellobiohydrolase activity of enzymes in *Fusarium oxysporum*)	[11,12]
Rice straw	H_2_O, 220 °C, 52 min and pH 7.0, in a 1 L Parr 4525 reactor, at 10% solid loading (wet weight)	Cellulase activity quantified by determining the required enzyme to release one µM of 4-methylumbelliferone/min. The enzyme loading (41.7 U/mL) was used.	Fractionation using centrifugal partition chromatography of hot water hydrolysates (from rice straw) exhibits a more efficient separation method than inhibiting different chemical classes using exo-cellulase (derived from the prehydrolysate). Phenolics derived from rice straw subjected to potent inhibition to cellulase enzymes followed by furans and acetic acid.	[13]
Corncob	Carbon-based solid (C-SO3H) acid catalyst/H_2_O at 120–160 °C for 4–6 h	Cellulase loadings of 20 and 40 FPU/g for 1–96 h	Pretreatment resulted in direct xylose released (78.1%) from corncob. Enzymatic hydrolysis of prehydrolysate yielded up to 91.6% in 48 h.	[14]
Switchgrass (Energy crop, *Panicum virgatum*)	Liquid hot water: 15% *w*/*w* solid content at 200 °C for 5 min	Cellulase in a ratio of 50 mg_protein_/g_glucan_ (20 FPU/g_glucan_), at the pH (4, 5, 6, and 7 for solid contents, 4.8 for 15%, and 5.5 for 20%).	Achieving high glucose yields and hydrolysis efficiencies recorded for high solid content (with high enzyme dosage)	[15]
Bamboo(*Dendrocalamus giganteus* Munro)	Non-isothermally pre-treated with hot H_2_O at 140–200 °C for different times (10–120 min)	Cellulase loading at 14.5 FPU/g	Maximum glucose yield was achieved (75.7%).	[16]
Residual poplar branches (Hardwood, 20 years old *Populus deltoides* trees)	Deionized H*_2_*O (liquid/solid ratio = 15) heating rate (~7 °C/min) to temperatures (170–220 °C), with reaction times (15–180 min)	Cellulase (60 FPU/g)	Prolong HT pretreatment at 170 °C (90 min) to 220 °C (15 min) showed improvement in glucose recovery (31.2 wt% versus 43.6 wt%)	[17]
Grapevine pruning (Hardwood, *Vitis vinifera*)	Deionized H*_2_*O (liquid/solid ratio = 15) heating rate (~7 °C/min) to temperatures (170–220 °C), with reaction times (15–180 min)	Cellulase (60 FPU/g)	Prolong HT pretreatment at 170 °C (90 min) to 220 °C (15 min) showed modest improvement in glucose recovery (54.0 wt% versus 63.0 wt%)	[17]
Pine tree sawdust (Softwood, *Pinus sylvestris*))	Deionized H*_2_*O (liquid/solid ratio = 15) heating rate (~7 °C/min) to temperatures (170–220 °C), with reaction times (15–180 min)	Cellulase (60 FPU/g)	Prolong HT pretreatment at 170 °C (90 min) to 220 °C (15 min) showed modest improvement in glucose recovery (21.6 wt% versus 22.6 wt%)	[17]
*Eucalyptus grandis*	Two-step liquid hot water pretreatment: (a) 5% *w*/*v* cellulosic enriched biomass in H_2_O at 180–200 °C for 10–60 min (b) Addition of H_2_O (5% *w*/*v* of dry solids) to the collected hydrolysate, heated (180–240 °C)	Cellulase -loading amount 40 FPU/g dry solid	Pretreatment of cellulose is highly dependent on reaction temperature rather than time. “*72 h digestibility of samples treated at 180 °C for 20 min was 72.8% (increased to 97.2% after the II^nd^ pretreatment at 240 °C). The slightest improvement in enzymatic digestibility from 81.2% to 86.6% was recorded when II^nd^ pretreatment reaction time was prolonged from 20 to 60 min at 200 °C.*”	[18]
Eucalyptus bark (*Cryptomeria japonica*)	CO_2_ ^a^/H_2_O at 175–200 °C for 4 h	Enzyme Mixture: Cellulases from *Trichoderma reesei*, pectinases from *Aspergillus niger*, and *α*-amylase from *Aspergillus oryzae.* Dosage of each enzyme = 1.25 U/mg	Glucose yield achieved about 80% (as an *α*-cellulose) during HT treatment.	[19]
Poplar wood	SO_2_ steam explosion	Cellulase Loading of 15 FPU/g Glucan	Glucose yields (74.3%) achieved. Total sugar yield (Glucose + Xylose = 95.9%)	[20]

^a^ act as an acid catalyst.

**Table 3 polymers-15-03671-t003:** The qualitative yields of glucose and xylose from stage 1 (pretreatment) and subsequent stage 2 (enzymatic process), where enzyme loading (30 mg/g glucan in the pre-washed untreated Dacotah switchgrass) was used (data from Wyman et al. [24]).

Pretreatments	Xylose Yields			Glucose Yields			Total Sugars		
↓/Stages	I	II	Total	I	II	Total	I	II	Total
Untreated	NA	1.9	1.9	NA	8.4	8.4	NA	10.3	10.3
Maximum			39.4			60.6			100
Dil. H_2_SO_4_	29.3	3.4	32.6	4.3	42.2	46.5	33.6	45.6	79.2
Sulfur dioxide steam explosion	28.7	3.2	31.9	3.0	48.3	51.4	31.7	51.5	83.2
Liquid hot water	25.9	5.3	31.3	4.1	47.3	51.4	30	52.6	82.6
Lime	13.6	22.4	36	0.9	54	54.9	14.5	76.4	90.9
Soaking in aqueous ammonia	9.5	17.8	27.3	0.2	39.8	40	9.7	57.6	69.2
Ammonia fiber expansion	11.1	25.6	36.7	0.8	47.1	47.9	11.9	72.7	84.6

**Table 4 polymers-15-03671-t004:** The conversion yield from cellulose using various catalysts under visible light.

	Light-On			Light-Off		
Catalysts	Glucose	HMF	*Total Conversion*	Glucose	HMF	*Total Conversion*
Au-HYT	48.1	10.6	*58.7*	15.5	0	*15.5*
Au-YT	30.2	2.6	*32.8*	10.2	0	*10.2*
HYT	14	10.1	*24.1*	11.1	7.5	*18.6*
YT	24.9	5.7	*30.6*	19.6	4.6	*24.2*
HY	20.2	19.1	*39.3*	18.9	17.1	*36*
HT	14.9	5.7	*20.6*	10.6	4.9	*15.5*

## Data Availability

Not applicable.

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
