# Peer review of "Light-Driven Depolymerization of Cellulosic Biomass into Hydrocarbons"

_polymers, 2023, doi:10.3390/polym15183671_

Round 1
Reviewer 1 Report
This review provides an interesting summarization of recent work on photocatalytic conversion of cellulosic biomass into hydrocarbons. The common cellulose depolymerization approaches have been introduced and their disadvantages are discussed. The recent advances on the photocatalytic conversion of cellulose have been critically reviewed. However, some aspects can be further improved. It is recommended for publication with addressing the following issues.
1. As a review paper, the scale of topic should be not too narrow, otherwise, only quite limited research can support the review and it is only appealing to less people. The photocatalytic conversion of cellulose into hydrocarbons is not a hot research topic currently, however, photocatalytic conversion of cellulose into hydrogen is more attractive. So it is suggested to include a parallel section for hydrogen production with C5-C6 compounds production. In the hydrogen production process, the C5-C6 compounds can be firstly produced, but would be consumed by the oxidized species like holes or hydroxyl radicals. The two aims are interconnected.
2. Another challenge for photocatalytic conversion of cellulose in section 2 is that how to inhibit the further oxidation of generated carbohydrates into CO2 by photocatalysis, without solving this key issue, the carbohydrate yield would not be high, so the strategy on this point could be provided.
3. Given that the review is focused on the photocatalysis conversion, the enzymatic or acid/base treatment could not be discussed in detail in section 3.1 and 3.2.
4. The topic is not precise, “Depolymerization of cellulosic biomass into hydrocarbons by photocatalysis“ is better.
5. More related literature should be included in the review.
Author Response
The authors thank the reviewer for providing valuable comments.
This review provides an interesting summarization of recent work on photocatalytic conversion of cellulosic biomass into hydrocarbons. The common cellulose depolymerization approaches have been introduced and their disadvantages are discussed. The recent advances on the photocatalytic conversion of cellulose have been critically reviewed. However, some aspects can be further improved. It is recommended for publication with addressing the following issues.
- As a review paper, the scale of topic should be not too narrow, otherwise, only quite limited research can support the review and it is only appealing to less people. The photocatalytic conversion of cellulose into hydrocarbons is not a hot research topic currently, however, photocatalytic conversion of cellulose into hydrogen is more attractive. So it is suggested to include a parallel section for hydrogen production with C5-C6 compounds production. In the hydrogen production process, the C5-C6 compounds can be firstly produced, but would be consumed by the oxidized species like holes or hydroxyl radicals. The two aims are interconnected.
Response: A section dedicated to Biomass derived Hydorgen production is added, just before the conclusion part.
- Another challenge for photocatalytic conversion of cellulose in section 2 is that how to inhibit the further oxidation of generated carbohydrates into CO2 by photocatalysis, without solving this key issue, the carbohydrate yield would not be high, so the strategy on this point could be provided.
Response: Few strategies (which we found in literature search) are included in the Main body.
- Given that the review is focused on the photocatalysis conversion, the enzymatic or acid/base treatment could not be discussed in detail in section 3.1 and 3.2.
Response: We revised accordingly.
- The topic is not precise, “Depolymerization of cellulosic biomass into hydrocarbons by photocatalysis“ is better.
Response: We revised the title with a more suitable one
- More related literature should be included in the review.
Response: More studies have been added to the current manuscript.

Reviewer 2 Report
This manuscript provides a detailed description of various current treatment methods for lignocellulosic biomass, focusing on the importance of photocatalysis in the treatment of lignocellulosic biomass, with a detailed description of the reaction mechanism. I believe the manuscript can be accepted with appropriate revisions.
1. The quality of illustrations in the manuscript needs to be improved, some illustrations are blurred such as Fig.1, Fig.3
2. There is relatively little information on enzyme treatment, and the explanation of the mechanism of enzyme treatment of biomass is not very clear.
3. The title of this article is "Hydrolysis of Cellulosic Biomass into Hydrocarbons using Light Energy", in which enzyme catalysis and acid treatment are also introduced, what are the advantages of photocatalytic treatment compared to these two catalytic methods? What are the advantages of photocatalytic treatment compared to these two catalytic methods? Does light energy play a role in enzyme catalysis and acid treatment, and if not, why do you compare these two catalytic methods with photocatalysis?
4. The introduction of photocatalysis in section 3.3 is too lengthy, and it is suggested to set subheadings to analyze it so that it is more convenient for readers to read and understand. The basic theory of activation energy is suggested to be placed before the introduction of various methods.
Minor editing of English language required
Author Response
The authors thank the reviewer for providing valuable comments.
This manuscript provides a detailed description of various current treatment methods for lignocellulosic biomass, focusing on the importance of photocatalysis in the treatment of lignocellulosic biomass, with a detailed description of the reaction mechanism. I believe the manuscript can be accepted with appropriate revisions.
- The quality of illustrations in the manuscript needs to be improved, some illustrations are blurred such as Fig.1, Fig.3
Response: Figure 1 and Figure 3 are redrawn again.
- There is relatively little information on enzyme treatment, and the explanation of the mechanism of enzyme treatment of biomass is not very clear.
Response: A new section added to the enzyme hydrolysis with more explanation.
- The title of this article is "Hydrolysis of Cellulosic Biomass into Hydrocarbons using Light Energy", in which enzyme catalysis and acid treatment are also introduced, what are the advantages of photocatalytic treatment compared to these two catalytic methods? What are the advantages of photocatalytic treatment compared to these two catalytic methods? Does light energy play a role in enzyme catalysis and acid treatment, and if not, why do you compare these two catalytic methods with photocatalysis?
Response: Key points and description are added in various sections.
- The introduction of photocatalysis in section 3.3 is too lengthy, and it is suggested to set subheadings to analyze it so that it is more convenient for readers to read and understand. The basic theory of activation energy is suggested to be placed before the introduction of various methods.
Response: Points related to mechanism are added.

Reviewer 3 Report
The review "Hydrolysis of Cellulosic Biomass into Hydrocarbons using Light Energy", dedicated to various methods of hydrolysis of cellulose-containing biomass in C5 and C6 organic compounds (in the name of hydrocarbons), includes an analysis of 45 sources of information, among which there is a single reference from 2023. The positive aspects of the review include an attempt to conduct a comparative analysis of enzymatic catalysis, chemical catalysis, and photocatalysis used for cellulose-containing raw materials, as well as the quality of the author's drawings, which, according to the authors, visualize the processes under discussion. The avoidable shortcomings of the review include the use of the cellulose formula in the drawings, the monomeric unit of which is cellobiose instead of glucose, criticism of enzymatic hydrolysis due to the "high cost" of enzyme preparations, as well as a distortion of the fundamental concept that in this case only the hydrolysis of cellulose and hemicelluloses is discussed. , and not all cellulose-containing biomass. Separately, I note that the above sources of information do not confirm the idea of the demand for photocatalysis or the actively developing direction of biomass processing.
Specific remarks:
1. Provide evidence (articles and reviews 2020-2023) that enzymatic hydrolysis is used in cases of efficient pre-treatment, as well as confirmation that the prices of enzymes for these purposes have decreased so much in recent years that there are productions based on biocatalysis. In my opinion, Finland does not have such industries.
2. Emphasize the advantage of chemical hydrolysis of cellulose-containing raw materials: this process does not require chemical pre-treatment to remove lignin.
3. Justify the chemical aspects of the application of photocatalysis not on pure cellulose or holocellulose, but on cellulose-containing raw materials with all components of the natural matrix.
5. Change the list of references, updating it with modern references.
6. After that, edit the abstract, excluding misinformation of readers about the advantages and disadvantages of enzymatic and chemical hydrolysis.
7. Proofread text and avoid terms that confuse readers, such as whether chemical hydrolysis always = acid hydrolysis.
Author Response
The authors thank the reviewer for providing valuable comments.
The review "Hydrolysis of Cellulosic Biomass into Hydrocarbons using Light Energy", dedicated to various methods of hydrolysis of cellulose-containing biomass in C5 and C6 organic compounds (in the name of hydrocarbons), includes an analysis of 45 sources of information, among which there is a single reference from 2023. The positive aspects of the review include an attempt to conduct a comparative analysis of enzymatic catalysis, chemical catalysis, and photocatalysis used for cellulose-containing raw materials, as well as the quality of the author's drawings, which, according to the authors, visualize the processes under discussion. The avoidable shortcomings of the review include the use of the cellulose formula in the drawings, the monomeric unit of which is cellobiose instead of glucose, criticism of enzymatic hydrolysis due to the "high cost" of enzyme preparations, as well as a distortion of the fundamental concept that in this case only the hydrolysis of cellulose and hemicelluloses is discussed. , and not all cellulose-containing biomass. Separately, I note that the above sources of information do not confirm the idea of the demand for photocatalysis or the actively developing direction of biomass processing.
Specific remarks:
- Provide evidence (articles and reviews 2020-2023) that enzymatic hydrolysis is used in cases of efficient pre-treatment, as well as confirmation that the prices of enzymes for these purposes have decreased so much in recent years that there are productions based on biocatalysis. In my opinion, Finland does not have such industries.
Response: Studies involving technoeconomical evaluation of enzymes, limitations, and newer studies are now added to this section.
- Emphasize the advantage of chemical hydrolysis of cellulose-containing raw materials: this process does not require chemical pre-treatment to remove lignin.
Response: We added information accordingly.
- Justify the chemical aspects of the application of photocatalysis not on pure cellulose or holocellulose, but on cellulose-containing raw materials with all components of the natural matrix.
Response: Revised accordingly.
- Change the list of references, updating it with modern references.
Response: We added more information and studies.
- After that, edit the abstract, excluding misinformation of readers about the advantages and disadvantages of enzymatic and chemical hydrolysis.
Response: Revised accordingly.
- Proofread text and avoid terms that confuse readers, such as whether chemical hydrolysis always = acid hydrolysis.
Response: Revised accordingly.

Round 2
Reviewer 2 Report
Accept in present form
Moderate editing of English language required
Reviewer 3 Report
The review, after peer review, has changed its title and has undergone a lot of changes, including an increase in the number of cited literature, which is very positive. It is clear to me that the authors have tried to respond to my comments, in particular, the section on enzymatic hydrolysis has been significantly expanded, including a new table. But at the same time, the authors used publications from 2003-2019 and only one publication in 2023. Unfortunately, the abstract and introduction remained the same in the article. I am sure that readers will understand that "photocatalytic conversion of cellulosic biomass" leads to results only when photocatalysis is applied directly to cellulose and hemicellulose previously isolated from biomass by non-photocatalytic methods. I decide to "publish as presented".